# Noise as Medicine: The Role of Microbial and Electrical Noise in Restoring Neuroimmune Tolerance Through Stochastic Resonance

**DOI:** 10.3390/neurosci6040118

**Published:** 2025-11-18

**Authors:** Eneidy Piña Mojica, Joao Victor Ribeiro, Felipe Fregni

**Affiliations:** Neuromodulation Center and Center for Clinical Research Learning, Spaulding Rehabilitation Hospital, Harvard Medical School, Boston, MA 02115, USA; epina7@mgb.org (E.P.M.); jvgomes@mgb.org (J.V.R.)

**Keywords:** stochastic resonance, microbial noise, hygiene hypothesis, neuroimmune disorders, vagus nerve stimulation, T-cell receptor, NF-κB, cytokine quorum sensing, fibromyalgia, multiple sclerosis

## Abstract

The rising prevalence of neuroimmune disorders such as multiple sclerosis and fibromyalgia has renewed interest in the hygiene hypothesis, which posits that reduced early-life microbial exposure deprives the immune system of formative “noise” that calibrates thresholds of tolerance. We extended this framework by introducing stochastic resonance (SR), a system phenomenon in which optimally tuned noise enhances weak-signal detection in nonlinear networks, as a potential surrogate for missing microbial variability. As electrical noise and subthreshold stimulation have been shown to modulate cortical excitability and enhance perception, microbial noise may be necessary for sustaining immune plasticity. Conversely, a lack of stimulation, whether microbial or electrical, can lead to maladaptive states characterized by dysregulated signaling and heightened vulnerability to chronic inflammation. Evidence from immunology highlights noise-aware processes, such as T-cell receptor proofreading, NF-κB pulsatility, and cytokine quorum sensing, all of which exploit stochastic fluctuations. Computational tumor–immune models similarly suggest that tuned noise can optimize immune surveillance. Clinical data from neuroscience demonstrate that subsensory electrical noise improves motor excitability and sensory perception, whereas vagus nerve stimulation modulates inflammatory pathways, underscoring translational feasibility. We propose that SR reframes noise from a biological error to a therapeutic resource capable of recalibrating dysregulated neuroimmune thresholds. This conceptual synthesis positions microbial and electrical noise as parallel modulators of tolerance and outlines testable predictions with translational potential for neuroimmune disorders.

## 1. Introduction

In recent decades, the incidence of chronic neuroimmune disorders, such as multiple sclerosis (MS) and fibromyalgia (FM), has risen markedly worldwide. This trend has increased academic interest in their poorly understood causes, particularly the role of early-life environmental influences on immune maturation and tolerance. A prominent explanation is the hygiene hypothesis, which was initially proposed to account for allergic disease but has now been extended to autoimmune and neuroimmune conditions, supported by converging epidemiological and mechanistic evidence [1,2]. According to this hypothesis, reduced microbial exposure, driven by urban living, widespread antibiotic use, modern delivery methods, and lifestyle changes, compromises immune training, weakens tolerance, and increases susceptibility to immune-mediated diseases [3,4,5,6].

A central concept within this framework is “microbial noise”, which is a steady, low-level stimulation provided by diverse microbial exposures that continuously modulate host immune signals. Such variability appears to be critical for calibrating immune thresholds and sustaining immunomodulatory pathways. In its absence, signaling may become dysregulated, predisposing to chronic inflammation and neuroimmune dysfunction [7,8,9].

From a systems perspective, microbial noise can be compared to stochastic resonance (SR), a phenomenon in which appropriately tuned random noise enhances weak signal detection in nonlinear systems [10]. In neuroscience, a comparable concept is “electrical noise,” where a subthreshold or random electrical fluctuations can amplify weak neural signals and modulate cortical excitability. Although still nascent in the context of immune signaling, SR offers a conceptual framework for understanding how background fluctuations, whether microbial, electrical, or intracellular, might optimize immune regulation.

This review asks a provocative question: Can stochastic resonance act as a functional analog of microbial “noise” and compensate for its loss in neuroimmune disorders? We present a hypothesis-generating framework that integrates immunology, neurophysiology, and systems biology. If validated, this perspective may open new therapeutic avenues for conditions characterized by impaired sensory and immune regulation, including fibromyalgia and multiple sclerosis.

## 2. The Hygiene Hypothesis & Immune System Training

The hygiene hypothesis, also referred to as the “old friends” hypothesis, was first articulated to explain the increasing prevalence of allergic diseases in industrialized countries, also this hypothesis has proposed that reduced microbial exposure in early life leads to an under-stimulated immune system [11], associated with widespread modern societal and medical practices [12].

Early and sustained contact with diverse microbial environments, particularly the gut, provides crucial molecular cues for immune development. Commensal microbes and their metabolites, such as short-chain fatty acids, promote the induction and maintenance of regulatory T cells (Tregs) and tolerogenic antigen-presenting cells, both of which are essential for suppressing excessive immune activation and preserving self-tolerance [7,13].

Animal models have reinforced these findings; germ-free or antibiotic-treated mice exhibit impaired Treg development, heightened pro-inflammatory responses, and increased susceptibility to immune-mediated diseases [14]. When early microbial “training” is absent or insufficient, immune programming skews toward pro-inflammatory phenotypes [7,15], predisposing individuals to allergic sensitization, inappropriate autoimmune activation, and chronic inflammation.

In essence, the hygiene hypothesis reflects more than an epidemiological pattern; it hypothesizes a mechanistic relationship between microbial variability, immune tolerance, and the developmental calibration of immune thresholds.

### Results Consequences of Missing Early-Life Immune Training

Reduced microbial exposure in early life deprives the immune system of the dynamic, low-grade stimulation required to discriminate between harmful and harmless stimuli. These microbial signals act as physiological “noise,” expanding the antigenic repertoire, enhancing Treg induction, and balancing cytokine networks toward appropriate activation [16,17,18].

In their absence, immune maturation can become rigid, hypervigilant, and prone to overreaction. Loss of antigenic diversity limits adaptive breadth, and reduced Treg function is associated with poor tolerance. Microbial deprivation can induce lasting epigenetic changes that may contribute to altered cytokine profiles in adulthood [8,9]. Disrupted microbial–immune communication may contribute to microglial activation, lowered sensory thresholds, hallmarks of FM, and related disorders [19,20] (Figure 1).

## 3. Linking Microbial Noise Loss to Neuroimmune Dysfunction

Loss of early-life microbial diversity and exposure, a central premise of the hygiene hypothesis, has been increasingly implicated in neuroimmune disorders. Two clinically distinct but mechanistically overlapping conditions, multiple sclerosis and fibromyalgia, illustrate how microbial deprivation can disrupt immune tolerance, destabilize neuroimmune networks, and foster chronic inflammation.

The shared thread in MS and FM is not only immune dysregulation, but also the loss of variability in microbial noise that contributes to fine-tuning immune thresholds and sensory processing.

### 3.1. Multiple Sclerosis

Multiple sclerosis (MS) is a prototypical autoimmune disorder of the central nervous system, characterized by inflammatory demyelination, neurodegeneration, and episodic or progressive neurological decline. Mounting evidence suggests that insufficient early-life microbial exposure contributes to a breakdown in immune tolerance, leading to autoreactive T cell responses against myelin antigens [21]. Epidemiological studies have revealed a consistently lower incidence of MS in rural and helminth-exposed populations, where microbial diversity and environmental contact are greater [22].

Strikingly, helminth-infected patients with MS demonstrate durable anti-inflammatory profiles marked by elevated interleukin-10 (IL-10), expansion of regulatory B cells, and enhanced TAM receptor signaling, which correlates with reduced disease activity [23,24,25].

This geographic patterning underscores that the immune architecture is not determined solely by genetics, geographic location, or diet, but also by the diversity and timing of microbial encounters. Disruption of the gut microbiota can compromise regulatory T cell induction and erode immune tolerance, creating a permissive environment for autoimmunity. The global rise of and epidemiological shifts in MS likely reflect, in part, a steady decline in beneficial microbial exposure, linking population trends directly to the gut–immune axis [1,22,26,27,28].

### 3.2. Fibromyalgia

FM is a chronic pain syndrome characterized by widespread musculoskeletal pain, fatigue, cognitive dysfunction, and heightened sensory sensitivity. While its pathophysiology remains incompletely defined, converging evidence implicates neuroinflammation and immune dysregulation, including elevated pro-inflammatory cytokines, such as IL-6 and IL-8, along with diminished anti-inflammatory signaling [29]. Compared with MS, evidence linking early-life microbiome alterations to FM is more limited and indirect.

Emerging perspectives suggest that inadequate early-life microbial exposure may predispose individuals to maladaptive pain processing and systemic immune imbalance [30]. Microbial deprivation may blunt the maturation of regulatory immune networks, biasing the system toward a pro-inflammatory phenotype that amplifies nociceptive signaling and disrupts central pain modulation.

This mechanistic framework also parallels other immune-mediated conditions, where gut–brain–immune axis dysregulation serves as a common pathogenic thread, which highlights the need to examine microbiome preservation and restoration not only as a preventive measure but also as a potential therapeutic adjunct. Such approaches can recalibrate immune tone, dampen neuroinflammation, and normalize sensory thresholds, offering a biologically grounded avenue for intervention.

## 4. What Is Stochastic Resonance (SR)?

Stochastic resonance is a phenomenon observed in nonlinear, threshold-controlled systems in which the addition of an optimal level of random noise enables weak, subthreshold signals to surpass activation thresholds. Rather than degrading the performance, this controlled noise increases the signal-to-noise ratio and enhances detection accuracy. For SR to occur, three conditions must be satisfied: the system must be nonlinear, the signal must be too weak to cross the threshold, and the noise must be tuned to an optimal intensity. In linear systems, noise simply masks information, but in nonlinear systems, such as neural circuits, sensory pathways, and immune signaling networks, noise can paradoxically enhance functions.

This effect has been repeatedly demonstrated in sensory physiology; [31] showed that subsensory mechanical vibration applied to the skin enhanced tactile perception in healthy adults, while ref. [32] found that low-level acoustic noise improved speech recognition in cochlear implant users, likely by increasing the responsiveness of auditory nerve fibers. In addition, in proprioceptive systems, SR-based interventions have yielded clinical benefits in neurological populations; ref. [33] reported that stochastic whole-body vibration improved postural stability in patients with multiple sclerosis, a finding reinforced by ref. [34] meta-analysis demonstrating consistent balance improvements across neurologically impaired cohorts. Similarly, ref. [35] found that SR delivered through shoe insoles improved gait and reduced postural sway in older adults with sensory loss, demonstrating that SR-based techniques can be used safely in humans.

Beyond sensory pathways, SR can also enhance computation inside nonlinear networks by adding a tuned level of noise, which can reduce input–output information loss and improve approximation accuracy, with performance peaking at an intermediate noise intensity [36].

Although the application of SR to the sensory and motor systems is well established, emerging theoretical and experimental studies suggest that its principles may extend to immune signaling. Ref. [37] proposed that T-cell receptor–MHC interactions can be modeled in the form of suprathreshold SR, wherein stochastic molecular fluctuations in receptor clustering enhance the detection of weak antigenic signals. In oncology models, stochastic perturbations have been shown to improve tumor–immune recognition without triggering pathological activation [38].

Just as immune tolerance depends on a precise balance between under- and over-activation, SR operates on a “Goldilocks” principle that suggests that too little noise leaves weak signals undetected, while too much noise overwhelms the system [18]. Early-life microbial exposures provide this optimal variability for immune calibration—training networks to discriminate against danger from safety and fine-tune thresholds for activation. SR offers a potential means of reintroducing this beneficial variability in a controlled, non-invasive manner, intending to restore healthy activation thresholds in both the sensory and immune domains [17,39,40]. Noise-assisted threshold-crossing and noise-aware immune signaling are summarized in Figure 2.

It is important to emphasize that SR is conceptualized here as a heuristic and system-level metaphor rather than a validated biological mechanism. When considered through the lens of a heuristic, it is understood that framing SR as a unifying model is more beneficial than framing it as a mechanistic claim about immune or neural function. Rather than serving as a mechanistic explanation, it should be viewed as a generative framework for developing and testing new hypotheses. Table 1 illustrates the conceptual parallels between stochastic resonance and immune processes.

## 5. Can Stochastic Resonance Replace Microbial “Noise”?

### 5.1. Theoretical Foundation: Noise as a Surrogate for Microbial Variability

According to the hygiene hypothesis framework, early-life microbial “noise” can calibrate immune thresholds, balancing defense and tolerance; in the absence of this stimulation, which might lead immune networks to develop into a hyperalert, maladaptive state that predisposes individuals to autoimmune, allergic, and neuroimmune disorders. Stochastic resonance may offer a way to recreate artificially, precisely, and in demand for missing stimulation.

In addition to individual receptor dynamics, immune collectives rely on noisy signals to coordinate responses. Adaptive thresholds in T cell populations depend on fluctuating IL-2 fields that implement quorum sensing, where variability in local cytokine concentrations provides critical contextual information for activation. This demonstrates that immune systems routinely exploit stochastic fluctuations to calibrate set points and balance responsiveness, reframing noise not as detrimental but as an essential component of immune regulation [41,42,43].

Just as microbial “noise” shapes immune tolerance in early life, SR could provide a physiologically meaningful substitute, restoring healthy thresholds in systems destabilized by environmental change or disease.

### 5.2. Biological Plausibility and Physiological Relevance

Stochastic resonance is a counterintuitive phenomenon in which the addition of an optimal level of random noise enhances the detection and transmission of weak subthreshold signals in nonlinear systems [10,44]. Crucially, the noise must be precisely tuned; too little fails to activate the signal, while too much overwhelms it. This “Goldilocks principle” has been demonstrated in sensory physiology and clinical studies, where subsensory noise improved tactile perception and balance control [31,45].

The same bell-shaped dose response has been demonstrated in noise-modulated neural networks, where the accuracy is maximized at a mid-range noise level and deteriorates with too little or too much noise [36]. Immune regulation exhibited striking parallels. Just as under- or over-stimulation destabilizes the sensory systems, insufficient immune input produces rigidity and poor tolerance, whereas excessive input risks pathological activation. Theoretical models suggest that stochastic fluctuations in TCR–MHC interactions sharpen antigen discrimination, effectively acting as adaptive noise that improves specificity [37].

T-cell antigen recognition integrates kinetic proofreading with stochastic contact times, allowing weak ligands to cross activation thresholds when supported by optimally noisy microdynamics [46]. Experimental evidence supports this framework: studies demonstrate that proofreading is distributed across multiple downstream steps, with binding lifetime fluctuations shaping signaling outcomes [47]. Complementary theoretical models have shown that stochastic variability enhances both sensitivity and specificity, enabling low-abundance agonists to trigger responses without global overactivation [48].

At the system level, noise enhances the information capacity. Information-theoretic analyses have revealed that dynamic, noise-aware signaling maximizes information transfer in biochemical networks, supporting the existence of an optimal noise regime for weak-signal detection [49,50]. NF-κB dynamics illustrate the following principle: inputs are encoded as digital pulses whose information capacity is maximized by fluctuation patterns, with different cell types uniquely interpreting these dynamics uniquely [51]. Cytokine-mediated quorum sensing provides another example, in which variability in IL-2 fields calibrates activation thresholds across T-cell populations [41,43].

Clinically, noise-like modulation is also possible. Patterned vagus nerve stimulation activates the inflammatory reflex, lowering cytokines such as TNF and IL-6 in humans, suggesting that external electrical variability can reset immune set points [52,53,54]. Together, these precedents show that the immune system does not merely tolerate noise, but actively harnesses it for discrimination, coordination, and homeostasis. Framing SR as an explanatory scaffold unifies these disparate mechanisms under a common systems principle without overclaiming experimental proof. In this light, SR serves not as a speculative novelty, but as a bridge linking established noise utilization strategies in immunity to future translational approaches in neuroimmune modulation.

### 5.3. Subthreshold and Weak Stimuli as Powerful Modulators

Studies have demonstrated that weak stimulation, and even subthreshold stimulation, can significantly affect brain excitability. The absence of stimulation, however, can have negative consequences [55]. These findings underscore the paradoxical power of “weak signals” to shape neural dynamics, suggesting that the nervous system is sensitive to inputs near the threshold of perception.

Evidence from cross-modal interactions shows that subthreshold somatosensory stimulation paired with subthreshold visual cortex stimulation via transcranial magnetic stimulation. Although neither stimulus alone reached perceptual awareness, their spatiotemporally congruent combination generated a reliable phosphene perception. This finding demonstrated that subthreshold inputs could summate across modalities to induce behaviorally significant percepts, reflecting the underlying anatomical connectivity and attentional modulation mechanisms [56].

Expanding on this principle, another study examined the effects of visuospatial attention on near-threshold somatosensory stimuli. Using cross-modal cueing paradigms, the authors found that attention significantly improved discrimination and conscious detection of weak tactile stimuli. Reaction times were shortened, and response bias shifted toward less conservative reporting, highlighting how cognitive factors can amplify the salience of near-threshold inputs [57].

A third line of evidence comes from the domain of pain modulation and how a lack of stimuli may lead to higher perception of pain. In an experimental trial, different somatosensory behavioral tasks ranging from passive sensory input to learning-based tasks with visual feedback have been shown to modulate pain thresholds and cortical excitability. Specifically, targeted sensory stimulation increased pain thresholds (indicating less pain) in the stimulated hand and reduced cortical excitability as measured by TMS. Conversely, the absence of such stimulation lowered the thresholds (indicating more pain), reinforcing the idea that a lack of weak inputs can have adverse effects on sensory processing [58].

Taken together, these studies converge on a central insight that weak and subthreshold signals are not trivial or inert. Instead, they can recalibrate cortical excitability, bias perceptual decision making, and modulate pain sensitivity. These findings provide experimental support for the broader hypothesis that stochastic resonance and weak signal amplification mechanisms are fundamental for brain function. In the context of this paper, they extend the “Noise as Medicine” framework by demonstrating that near-threshold inputs can restore or enhance neural responsiveness, much like microbial variability shapes immune tolerance.

## 6. Computational Immunology & Tumor Control Models

Recent computational models have reinforced the potential of SR-like mechanisms to act as immune modulators. In tumor–immune interaction simulations, elevated noise intensity, especially when coupled with prolonged exposure, has been shown to significantly suppress tumor cell growth, suggesting noise-dependent control of immune surveillance [38]. White noise inputs combined with optimal control strategies, such as pulsed chemotherapy, have also been found to stabilize tumor–immune dynamics and promote tumor extinction. Together with tumor–immune models, these findings motivate dose-controlled SR protocols and prespecified performance metrics (e.g., information transfer and error rates) analogous to those used in noise-modulated neural networks [36].

Similarly, ref. [59] used stochastic simulations of effector and regulatory T cell networks to show that internal noise alone, without changes in antigenic input, can induce transitions between immune tolerance, oscillatory behavior, and runaway activation. These findings reinforce that stochastic fluctuations in small immune populations can dynamically modulate system thresholds, serving as internal sources of functional variability akin to stochastic resonance. The use of linear noise approximation and effective stability analysis offers rigorous, biologically grounded evidence for the plausibility of noise-induced immune-state transitions.

At the cellular level, ref. [37] proposes that the constant mosaic of selfpeptide–MHC complexes constitute “athermal noise” that amplifies weak non-self signals in T-cell receptors, effectively tuning detection thresholds via a noise-assisted mechanism. This adaptive threshold stochastic resonance model suggests that TCRs dynamically adjust their activation threshold through intracellular feedback and receptor clustering, enabling the specific recognition of foreign antigens amid a noisy background of self-signals.

Stochastic control strategies to enhance antibody production further illustrate the feasibility of deliberately introducing noise to boost immune responsiveness, thereby pointing to a broader theoretical framework for SR-based immune tuning.

## 7. Translational Potential

Based on this precedent, SR-inspired strategies can be deployed at several practical “injection points” across physiological systems. Bioelectric approaches, such as transcutaneous or implanted vagus nerve stimulation (VNS), directly engage the cholinergic anti-inflammatory pathway and have been shown to down-regulate cytokines such as TNF and IL-6 in both preclinical and clinical studies [52,53,54].

Beyond neuromodulation, recent work in primary human cells shows that direct electrical stimulation can polarize macrophages toward an anti-inflammatory, pro-regenerative state and suppress LPS-induced activation, reinforcing the feasibility of SR-informed electromodulation as an immune-tuning strategy.

Mechanosensory interventions, including subsensory vibration that enhances proprioceptive function, offer a low-risk means of recalibrating afferent–autonomic coupling [31,35,45]. At the cellular level, microfluidic platforms could deliver dynamic “flicker” patterns of cytokines such as IL-2 or interferons, providing a controlled way to experimentally tune immune thresholds and test the principle that noise variability enhances immune discrimination [41,43]. To test whether the benefits arise from stochastic inputs rather than generic stimulation, future studies should run energy-matched head-to-head waveform controls. All waveforms will be RMS-matched with identical duty cycle, session length, and electrode/actuator geometry; stochasticity and spectra will be verified by power spectral density, sample entropy, and autocorrelation [10,44].

Holding the delivered energy constant, only stochastic spectra (white/pink/band-limited noise) produce a U-shaped noise–response curve and enable subthreshold rescue [44]. We further predicted greater information transfer (e.g., NF-κB pulse code capacity; IL-2 channels capacity) without global hyperactivation under stochastic, rather than deterministic, drive [49,50,51]. Electromodulation has been shown to shift primary human macrophages toward anti-inflammatory and pro-regenerative phenotypes. The SR-informed bioelectric framework and its neuroimmune targets are shown in Figure 3.

Together, these approaches illustrate how stochastic inputs can be harnessed to restore the balance across neural, immune, and sensory networks. If successfully translated, SR-based interventions could inaugurate a new class of cross-system therapies that bridge neurophysiology and immunology. By reinstating the beneficial variability eroded by modern environments, medical interventions, or chronic diseases, SR offers a biologically plausible, non-pharmacological strategy to enhance resilience and adaptive capacity across both sensory and immune domains [37,38].

## 8. Limitations

Although multiple immune processes, such as NF-κB pulsatility, TCR proofreading, and cytokine quorum sensing, have already demonstrated noise-based dynamics, direct experimental evidence of stochastic resonance in immune signaling remains limited. Therefore, our synthesis is conceptual, and hypothesis driven. Second, the application of SR to immunity remains emergent and speculative and requires rigorous mechanistic testing to establish its validity. Therefore, experimental validation is essential.

Testable directions may include integrating microfluidic studies introducing dynamic cytokine “flicker” to probe immune threshold tuning, controlled noise inputs in TCR signaling assays to map antigen discrimination, and clinical translation through patterned vagus nerve or mechanosensory stimulation trials. These strategies will help determine whether SR can move from an analogy to an actionable framework.

### State of Evidence: What We Know and What Is Missing

However, direct demonstrations of SR in human immune signaling are lacking. At present, the strongest evidence comes from the following adjacent domains. Multiple immune processes can be used to exploit functional fluctuations. NF-κB encodes information on pulsatile dynamics, the channel capacity of which is shaped by noise [49,50,51]. T cell antigen discrimination relies on kinetic proofreading coupled with stochastic contact times, enabling threshold crossing by weak ligands [46]. Cytokine-mediated quorum sensing uses variability in IL-2 fields to calibrate collective activation thresholds [41,43]. This rationale is consistent with computational results showing that tuned noise improves performance even when the desired input–output mapping is inherently nonlinear by effectively smoothing hard thresholds to increase information transfer [36].

Humans benefit from noise-based input. In sensory physiology, subsensory vibration and acoustic noise reliably enhance perception and stability in humans [31,32,35,45]. In neuroimmune contexts, vagus nerve stimulation engages the inflammatory reflex and suppresses pro-inflammatory cytokines such as TNF and IL-6 [52,53,54], demonstrating that external patterned “noise-like” inputs can reset immune set-points.

No study has systematically dosed noise in immune readouts, for example, titrating the intensity, spectrum, and duty cycle of stochastic inputs to generate the characteristic U-shaped SR response curve in cytokines, Treg induction, or NF-κB pulsatility. This gap highlights the need for formal experiments such as the proposed microfluidic cytokine flicker assay, which would directly test whether immune thresholds display SR-like dynamics.

## 9. Conclusions

Loss of early-life microbial diversity deprives developing immune networks of formative “noise,” narrowing repertoires, weakening tolerance, and fostering maladaptive thresholds across immune and neural systems. Stochastic resonance reframes noise not as a defect but as a resource; in nonlinear thresholded biology, carefully tuned variability can amplify weak but meaningful signals while suppressing spurious activation. Evidence spanning sensory physiology, immune theory, and computational tumor–immune models converge on the same principle: there exists an optimal noise regime that restores sensitivity without provoking instability. Importantly, this optimal-noise behavior also holds in nonlinear networks, where tuned noise improves the mapping fidelity rather than merely approaching a linear bound [36].

Positioning SR as a functional analog of microbial noise offers a unifying system explanation and a translational path. Mechanistically, immune circuits that rely on kinetic proofreading, temporal decoding, and cytokine-mediated quorum sensing are inherently poised to benefit from the well-shaped fluctuations. Clinically, patterned bioelectric inputs, such as vagus nerve stimulation and low-level mechanosensory noise, modulate physiological set points, suggesting that noise-informed interventions are feasible and can be delivered noninvasively.

The key challenge is precision: noise must be dosed, timed, and patterned into the dynamics of the targeted circuit. This requires mechanistic biomarkers and rigor in trial design. Rather than asking whether “noise works,” the next phase should identify who benefits, at what intensity, with which temporal statistics, and by what readouts.

## Figures and Tables

**Figure 1 neurosci-06-00118-f001:**
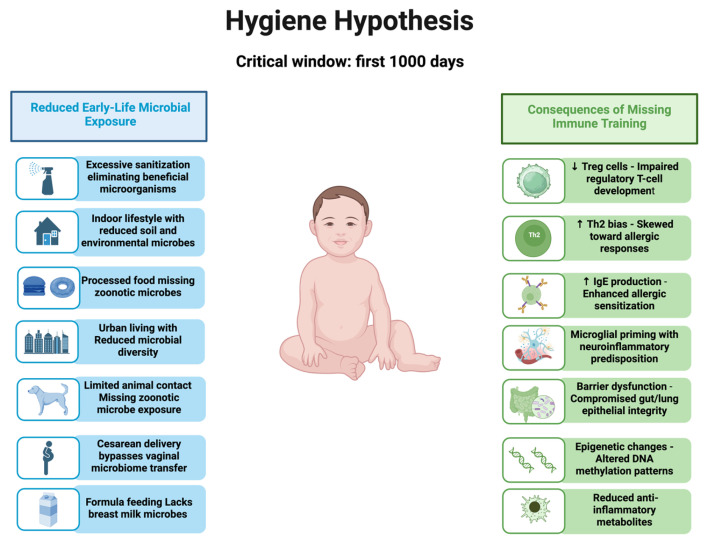
Consequences of reduced early-life microbial exposure as proposed in the hygiene hypothesis. Blue: causes of reduced microbial exposure during early life, and green: consequences of reduced early-life microbial exposure.

**Figure 2 neurosci-06-00118-f002:**
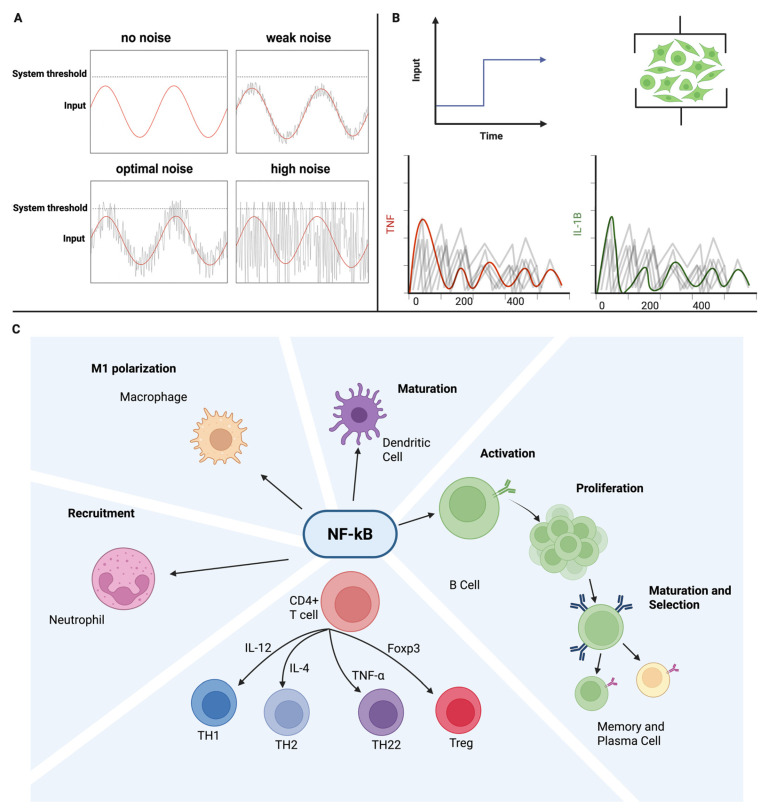
Noise-assisted signaling and NF-κB as a noise-aware immune hub. (**A**) Classic SR motif: a weak periodic input fails to cross threshold without noise; optimal noise enables reliable crossings; excessive noise yields false positives. (**B**) Input of SR to an immune cell population in order to elicit noisy and pulse-like cytokine outputs (e.g., TNF, IL-1β), illustrating information encoding in fluctuating dynamics. (**C**) NF-κB integrates noise-laden signals to orchestrate macrophage polarization, dendritic cell maturation, B-cell activation/proliferation, and CD4^+^ T-cell differentiation (TH1/TH2/TH22/Treg).

**Figure 3 neurosci-06-00118-f003:**
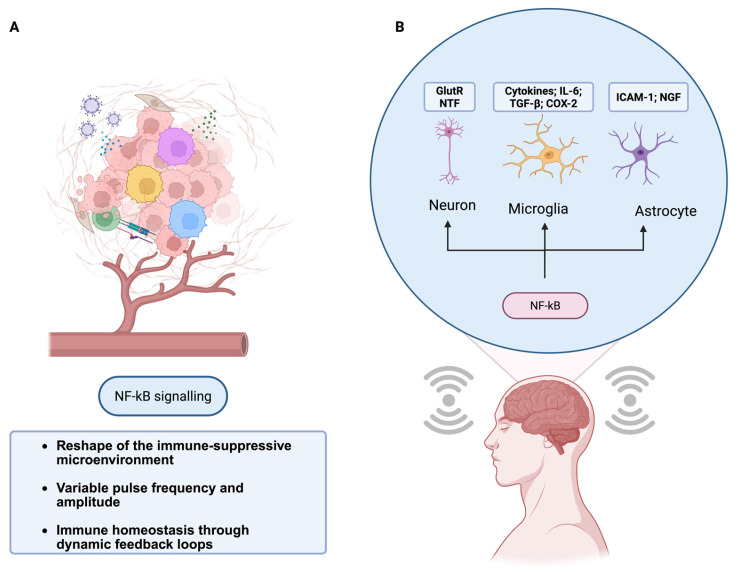
SR-informed bioelectric modulation of neuroimmune NF-κB dynamics. (**A**) Conceptual pathway by which patterned, noise-like inputs can reshape inflammatory microenvironments via NF-κB–coupled feedback, restoring homeostasis. (**B**) In the CNS, neuromodulatory inputs can influence NF-κB-linked neuron–microglia–astrocyte interactions, providing a rationale for noninvasive, noise-patterned stimulation in neuroimmune disorders.

**Table 1 neurosci-06-00118-t001:** Sensory SR and immune processes.

Dimension	Stochastic Resonance (SR)	Immune System Analog	Insight
1. Primary Function and Purpose	SR enhances the detection and transmission of weak stimuli by introducing an optimal level of noise. This noise acts as a facilitator that pushes subthreshold signals over the activation threshold of neurons, improving sensitivity and precision.	The immune system similarly relies on mild, continuous antigenic exposure (e.g., commensal microbes, dietary antigens, or vaccines) to calibrate its response thresholds. Controlled stimulation maintains immune vigilance while preventing overreaction to benign stimuli.	Both systems function as adaptive filters—they must distinguish relevant from irrelevant input. In each, a moderate amount of background variability supports robustness and adaptability.
2. Nature of the Input Signal	Inputs in sensory SR are typically weak, periodic, or subthreshold physical stimuli—such as faint vibrations, low-intensity light, or subtle auditory cues—that would be undetectable without noise-induced amplification.	The immune system’s inputs are antigenic signals: peptides, microbial molecules, or cytokine cues that can be either pathogenic or commensal in origin. Constant low-grade antigenic “noise” sustains immune readiness and tolerance.	In both cases, the system’s performance depends not only on the presence of input but on its distribution over time; sparse inputs require modulation by noise to maintain calibration.
3. Source and Role of Noise	Noise in sensory SR arises from electrical, synaptic, or environmental fluctuations. Rather than being detrimental, when tuned to the system’s threshold, it synchronizes neural firing and increases information transfer.	Immune “noise” emerges from microbial diversity, cytokine fluctuations, and stochastic gene expression in immune cells. These fluctuations help the system sample a broader response space, avoiding rigid or maladaptive immune states.	Noise acts as a training substrate for both systems—introducing variability that improves discrimination between meaningful and spurious signals, increasing flexibility.
4. The Optimal Zone	Too little noise leaves the system insensitive; too much noise leads to random or incoherent outputs. SR identifies a narrow optimal range of variability that maximizes signal-to-noise ratio and performance.	Immune systems also follow a Goldilocks principle: inadequate exposure (e.g., overly sterile environments) weakens tolerance development, whereas excessive exposure or dysbiosis induces hyperreactivity or chronic inflammation.	Both systems depend on maintaining an intermediate level of perturbation. This principle provides a framework for conceptualizing health as regulated variability, not static equilibrium.
5. System Output and Performance	The result of optimal SR is improved sensory discrimination, enhanced reaction timing, and increased perceptual acuity. The effect can be measured as improved signal detection probability or decreased error rates.	Properly “tuned” immune variability produces balanced immune activation: strong defense against pathogens without loss of tolerance to self or harmless antigens.	In both domains, performance is measured by the efficiency of correct detection and response. SR thus becomes a metaphor for adaptive optimization in complex biological systems.
6. Mechanisms of Regulation and Feedback	Neural SR involves dynamic feedback between sensory neurons, interneurons, and cortical circuits that adjust firing thresholds based on recent inputs and internal states. Neuromodulators (e.g., dopamine, serotonin) tune these dynamics.	The immune system’s feedback involves cytokine networks, regulatory T cells, and inhibitory checkpoints that modulate activation thresholds based on antigen load and inflammation history.	Both systems exhibit hierarchical feedback control—multi-scale loops maintain the system near a metastable state where responsiveness and stability coexist.
7. System-Level Integration	In the nervous system, SR may operate across scales—from ion-channel noise to population-level synchronization—linking micro and macro dynamics of perception and behavior.	In the immune system, variability propagates across levels—from stochastic receptor signaling to organism-level inflammatory tone—linking molecular noise to systemic homeostasis.	Both show scale-invariance of noise-benefit effects, suggesting that stochastic modulation may be a general organizing principle in biological adaptation.

## Data Availability

No new data were created or analyzed in this study. Data sharing is not applicable to this article.

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
