# Peer review of "Noise as Medicine: The Role of Microbial and Electrical Noise in Restoring Neuroimmune Tolerance Through Stochastic Resonance"

_neurosci, 2025, doi:10.3390/neurosci6040118_

Round 1

Reviewer 1 Report

Comments and Suggestions for Authors

The authors employed two different approaches to demonstrate the importance of a healthy brain and immune balance in lifestyle changes, which can be achieved through microbial exposure or mild electrical stimulation. It is a good starting point and will require further discovery studies to validate the approach.

The work is ambitious and well-structured, combining evidence from immunology, neurophysiology, systems biology, and computational modeling. It offers a conceptual bridge between microbial ecology and nonlinear systems theory, which is a very novel approach.

My main concern is the distinction between evidence, speculation, and analogy, and the need to tighten certain arguments to improve accessibility and translational clarity.

It would help to explicitly clarify that SR is being used as a heuristic or system-level metaphor, not yet a verified biological mechanism. Discussing this point, perhaps before the Discussion section that would help clearly distinguish between analogy and mechanism.

Adding a table summarizing sensory SR and immune processes would also help readers better understand the comparison.

The other approach is adding a conceptual figure or a schematic chart, such as a diagram linking microbial diversity, continued by immune noise, then stochastic resonance, and lastly ending with restored threshold regulation. I leave this up to the author's decision since I feel it will give more information on the article visually to understand the interesting concept.

Since this study is first its own may outline the specific experiments or measurable outcomes that could validate their hypothesis.

Reviewer 2 Report

Comments and Suggestions for Authors

The authors conducted a rather interesting parallel between "stochastic resonance" and "microbial noise." The review provides compelling evidence that "medical noise" influences on immunity and nerve system. At the beginning of the review, the authors argue that the more diverse the microbiome does the immunity for stronger. The main part of review is devoted to influence of different noises on immune and nerve systems. As such, the influence of microbiome diversity on the immune system remains purely hypothetical without serious evidence. This is essentially what the authors stated in their conclusion. Overall, I believe this manuscript, after correcting my comments, is suitable for publication in a NeuroSci journal.

Authors use two types of references: (1) citing authors and (2) using numbers. Please ensure that references are consistent in style.

Authors often use a citation style like "while (25) found" (i.e., without author’s name). I would recommend to use ordinary citing: for example, "Zeng et al (25) found"

Line 57-59. Microorganisms do influence immune system optimization, as demonstrated in numerous studies. Could the authors clarify whether there is indeed literature data suggesting that "electrical, or intracellular, mechanisms might optimize immune regulation?" Could the authors provide a references to these studies?

Line 181-189. There are no references in this paragraph. Is this the authors' statement? Or is this data from the literature? Please clarify and, if necessary, provide references.

Figure 2a – the top right and left figures are identical (“no noise” and “weak noise”). Is this correct?

Figure 2 is of poor quality; please replace it with a higher-resolution image. Perhaps you could enlarge the text in the images themselves.

Figure 2b – What do the cells in the square (upper right figure) mean? Add the explanation in the figure caption.

The authors define Stochastic Resonance in several places. These are Line 53-55, Line 150-152, and Line 216-218. Such varied interpretations of the term can be confusing to readers. I would recommend the authors define it only once.

Line 255: “One of the authors has conducted Several studies” – please set reference on this work

Line 260: “a seminal study on crossmodal interactions” – What work are Authors talking about? Please clarify.

Line 267: “another study examined” – What work are Authors talking about? Please set reference.

Line 288 “microbial variability” may be “microbial diversity”?

Figure 3a: there are no any description of different type of cell. Please set description of each cell type. And set “blood vessel”.

Reviewer 3 Report

Comments and Suggestions for Authors

This is a very comprehensive review introducing the general physical principle of stochastic resonance to immunology and its implications during individual development of immunity in childhood and to immunological reactions in adults. 

This novel interdisciplinary approach looks very fruitful and can lead to new therapeutic regimens.

The manuscript is designed and performed very well - no major flaws and no criticism from my side. 

Only one minor thing: In paragraph 5.3 the first sentence is lumped together from two intended sentences.

Round 2

Reviewer 1 Report

Comments and Suggestions for Authors

Thank you for the clarifications and revisions. The added prefatory paragraph clearly distinguishing SR as a heuristic framework and helps for readers. The added table contrasting sensory SR and immune processes is also a useful addition, enhancing clarity around the analogy. I understand the decision not to add an additional figure given the current heuristic nature of the model.

Overall, these revisions satisfactorily address my comments.